# Chlorogenic Acid Isomers Isolated from *Artemisia lavandulaefolia* Exhibit Anti-Rosacea Effects In Vitro

**DOI:** 10.3390/biomedicines10020463

**Published:** 2022-02-16

**Authors:** Kyung-Baeg Roh, Youngsu Jang, Eunae Cho, Deokhoon Park, Dae-Hyuk Kweon, Eunsun Jung

**Affiliations:** 1Biospectrum Life Science Institute, Yongin 16827, Korea; biosh@biospectrum.com (K.-B.R.); biogc@biospectrum.com (Y.J.); biozr@biospectrum.com (E.C.); pdh@biospectrum.com (D.P.); 2Department of Integrative Biotechnology, College of Biotechnology and Bioengineering, Sungkyunkwan University, Suwon 16419, Korea; dhkweon@skku.edu

**Keywords:** rosacea, *Artemisia lavandulaefolia*, chlorogenic acid isomers, kallikrein 5, cathelicidin

## Abstract

Rosacea is a chronic inflammatory disease affecting facial skin. It is associated with immune and vascular dysfunction mediated via increased expression and activity of cathelicidin and kallikrein 5 (KLK5), a serine protease of stratum corneum. Therefore, KLK5 inhibitors are considered as therapeutic agents for improving the underlying pathophysiology and clinical manifestation of rosacea. Here, we isolated the active constituents of *Artemisia lavandulaefolia* (*A. lavandulaefolia*) and investigated their inhibitory effect on KLK5 protease activity. Using bioassay-guided isolation, two bioactive compounds including chlorogenic acid isomers, 3,5-dicaffeoylquinic acid (isochlorogenic acid A) (1), and 4,5-dicaffeoylquinic acid (isochlorogenic acid C) (2) were isolated from *A. lavandulaefolia*. In this study, we evaluated the effects of isochlorogenic acids A and C on dysregulation of vascular and immune responses to rosacea, and elucidated their molecular mechanisms of action. The two chlorogenic acid isomers inhibit KLK5 protease activity, leading to reduced conversion of inactive cathelicidin into active LL-37. This inhibition of LL-37 production by isochlorogenic acids A and C reveals the efficacy of suppressing the expression of inflammatory mediators induced by LL-37 in immune cells such as macrophages and mast cells. In addition, both isomers of chlorogenic acid directly inhibited the proliferation and migration of vascular endothelial cells induced by LL-37.

## 1. Introduction

Nearly 30 members of the Cathelicidin family have been identified in mammalian species. In humans, *CAMP* is the only cathelicidin gene encoding the inactive precursor protein, human cationic antibacterial protein of 18 kDa (hCAP18), which is the highly conserved cathelin domain and antimicrobial c-terminal domain [1]. The antimicrobial peptide LL-37 is generated via proteolytic processing of cathelicidin hCAP18 by stratum corneum serine protease, kallikrein 5 (KLK5), in the human epidermis, and mediates the innate defense mechanism against invading pathogens via direct lethal effect against microbes, resulting in immune cell function [2]. This function serves as a signal initiating adaptive immune defense. Recent findings suggest that proteolytically generated LL-37 promotes leukocyte chemotaxis, angiogenesis, and inflammasome activation [3,4,5]. Rosacea lesions expressed elevated KLK5 in the basal layer of epidermis along with cathelicidin, in contrast to healthy skin, expressing reduced levels of cathelicidin and KLK5 superficially [6]. Increased KLK5 activity in rosacea generates LL-37 by cleaving cathelicidin, which triggered cutaneous inflammation and erythema.

Rosacea is characterized by various skin symptoms such as flushing, erythema, and telangiectasia due to neurovascular dysregulation in association with skin inflammatory response and is attributed to abnormalities of the innate immune system [7]. Various extracellular cytokines, chemokines, and growth factors have been shown to increase angiogenesis and chemotactic migration in vascular endothelial cells [8,9,10]. Among the various factors secreted extracellularly, LL-37 has been reported to modulate physiological activities such as chemotactic migration and angiogenesis in human umbilical vein endothelial cells (HUVECs), which are activated via G protein-coupled formyl peptide receptor-like 1 (FPRL1) signaling pathway [11]. Since the activation of vascular endothelial cells plays a central role in the regulation of angiogenesis and vascular inflammation, the identification of molecules that inhibit the G protein-coupled receptor (GPCR)-mediated signaling of vascular endothelial cells is important for the development of therapeutic agents targeting vascular diseases such as rosacea.

Mast cells are found at the boundary between organs, tissues, and external environments such as skin, gastrointestinal tract, and epithelium, and differ in inflammatory response and release of inflammatory mediators depending on the tissue or organ location [12,13]. Mast cells are activated by other immune cells, including macrophages, neutrophils, and T cells, and are involved in the pathogenesis of rosacea via release of preformed inflammatory mediators such as pro-inflammatory cytokines, chemokines and proteases that induce skin inflammation [14,15,16]. Recently, mast cells have emerged as a key factor in LL-37-induced rosacea [17,18]. Therefore, the inflammation in rosacea can be modulated by inhibiting the generation of LL-37 via inhibition of KLK5 activity, or by directly blocking the release of inflammatory mediators induced by LL-37 in mast cells.

Many active ingredients currently used as pharmaceuticals have been isolated from natural products, many of which are based on their use in traditional medicine. Herbal plants contain secondary metabolites such as alkaloids, flavonoids, terpenoids, saponins, glycosides, phenolic compounds, and tannins as active ingredients, and these are responsible for various biological activities [19,20].

*Artemisia lavandulaefolia* (*A. lavandulaefolia*) is a perennial belonging to the Asteraceae family and has been used in traditional medicine. The chemical composition of *A. lavandulaefolia* has been characterized in several studies and whole plant extracts or essential oil containing active components or secondary metabolites exhibit a wide range of bioactivity, including anti-microbial, anti-fungal, anti-inflammatory, anti-angiogenic, and insecticidal [21,22,23,24,25,26]. However, the efficacy of *A. lavandulaefolia* in rosacea has yet to be evaluated or reported. The bioactive compounds of *A. lavandulaefolia* with therapeutic effectiveness in rosacea have yet to be identified, and their mechanisms of action have yet to be elucidated.

In this study, we isolated active compounds that inhibit KLK5 activity from *A. lavandulaefolia* via bioassay-guided fractionation. In addition, we elucidated the structure of the isolated compounds and evaluated their anti-rosacea properties. The isolated compounds, chlorogenic acid isomers effectively inhibited the activity of KLK5, which regulates the generation of cathelicidin into the active form of LL-37. Our findings showed the effect of chlorogenic acid isomers on rosacea-related molecular targets, suggesting their potential as therapeutics for rosacea.

## 2. Materials and Methods

### 2.1. Cell Culture

Human epidermal keratinocytes (HEKn, Gibco, Waltham, MA, USA) were maintained in EpiLife medium (Gibco) supplemented with Human Keratinocyte Growth Supplement (HKGS, Gibco) at 37 °C in the presence of 5% CO_2_. THP-1 human monocytes were maintained in RPMI 1640 medium (WELGENE Inc., Gyeongsan, Korea) containing 10% fetal bovine serum (FBS, Sigma-Aldrich, St. Louis, MO, USA) and 1% penicillin/streptomycin (Gibco) at 37 °C under 5% CO_2_. Mouse mast cell line MC/9 was maintained in Dulbecco’s Modified Eagle Medium (DMEM), containing 10% T-Cell Culture Supplement with ConA (T-STIM, Corning, NY, USA), 10% FBS, 0.05 mM 2-mercaptoethanol (Sigma Aldrich), and 1% penicillin/streptomycin, at 37 °C, under 5% CO_2_. Human endothelial microvascular cell lines (HMEC-1, CRL-3243™, ATCC^®^ Manassas, VA, USA) were maintained in MCDB 131 (Gibco), containing 10 ng/mL EGF (Invitrogen, Carlsbad, CA, USA), 1 µg/mL hydrocortisone (Sigma-Aldrich, St. Louis, MO, USA), 10 mM glutamine (Gibco), and 10% FBS at 37 °C in the presence of 5% CO_2_.

### 2.2. Chemical Standards and Solvents

Isochlorogenic acid A, isochlorogenic acid B, and isochlorogenic acid C were obtained from ChemFaces (Wuhan, Hubei, China). Chlorogenic acid and trifluoroacetic acid (TFA, ≥99.0%) were purchased from Sigma-Aldrich. HPLC grade acetonitrile (≥99.9%) and methanol were purchased from J.T. Baker (Phillipsburg, NJ, USA).

### 2.3. Preparation of A. lavandulaefolia extract (ALE)

The aerial part of *A. lavandulaefolia* was obtained from the ChangHeung Co. (Namwon, Korea). *A. lavandulaefolia* was extracted with water at 90 °C for 3 h. The water extracts were evaporated and lyophilized to obtain completely dried ALE. The obtained extract was dissolved in deionized water for further experiments.

### 2.4. Preparation of Standard and ALE Solution

The isochlorogenic acid A and isochlorogenic acid C standard were dissolved in methanol (1.0 mg/mL) with an appropriate sonication, and ALE was dissolved in methanol (10.0 mg/mL) with an appropriate sonication. The standard and test solutions were filtered through a 0.22 μm syringe filter (SmartPor-II with universal hydrophilic polytetrafluoroethylene membrane, Sigma-Aldrich) prior to performance of the HPLC injection.

### 2.5. HPLC Analysis

ALE was quantitatively analyzed by HPLC system (Waters 2695, Waters Corp., Milford, MA, USA), equipped with a Waters 2996 Photodiode Array (PDA Detector). The Empower 2 software was used to control the analytical system and perform the data collection and processing. For HPLC-PDA was performed on a Luna C18(2) (4.6 × 250 mm, 5 μm) column reversed-phase column protected by a C18 guard column from Phenomenex, Inc. (Torrance, CA, USA). The sample injection volume was 10 μL. The signal was monitored at 254 nm. The elution system used for the HPLC-PDA assay was a binary high-pressure gradient elution system with mobile phase A (0.1% trifluoroacetic acid, TFA in H_2_O) and mobile phase B (acetonitrile), which was run with the gradient as follows: 0 min, 10% B; 15 min, 28% B; 40 min, 32% B; 45 min, 100% B; 50 min, 100% B; 55 min, 10% B; 60 min 10% B. The flow rate was 1.0 mL/min. Each analysis required 60 min including the re-equilibration time.

### 2.6. HPLC MS/MS Analysis

The HPLC-MS/MS systems consisted of a Ultimate3000 liquid chromatography system (Thermo Fisher Scientific, Waltham, MA, USA) coupled to a Triple TOF 5600+ (Sciex, Framingham, MA, USA) equipped with an electrospray ionization (ESI) interface. The chromatographic separation was undertaken with LC/MS equipped Waters CORTECS C18 (2.1 mm × 150 mm, 1.6 μm) (Waters Corp.). For the HPLC, 2 μL of the sample was injected with a flow rate of 300 μL/min with a temperature of column oven (40 °C). The mobile phases were 0.1% formic acid in H_2_O (A) and 0.1% formic acid in acetonitrile (B), and the following gradient was used: 5% B (0–1 min), 5–20% B (1–20 min), 20–100% B (20–45 min), 100% B (45–50 min), 100–5% B (50–51 min) and 5% B (51–60 min). The ESI parameters were set as follows: in negative ion mode in Continuum format, an ion spray voltage 4.5 kv, desolvation temperature of 500 °C, nebulizing gas 50 psi, heating gas 50 psi, curtain gas 25 psi. The spectra were recorded in the range of *m*/*z* 100–2000 for full scan MS and *m*/*z* 30–2000 for full scan MS/MS analysis.

### 2.7. KLK5 Protease Activity

KLK5 activity was assayed using recombinant human KLK5 proteins (rhKLK5, R&D systems, Minneapolis, MN, USA) with NaH_2_PO_4_ buffer (pH 8.0) at room temperature. KLK5 protease activity was measured by its ability to cleave the fluorogenic peptide substrate, Boc-V-P-R-AMC (R&D systems) in the kinetic assay. Briefly, recombinant human KLK5 was preincubated with ALE or the isolated compounds or leupeptin, a serine protease inhibitor for 10 min, followed by the addition of fluorogenic peptide substrate. Relative fluorescence units (RFUs) were measured at Ex 380 nm/Em 460 nm in kinetic mode for 5 min. Relative KLK5 activity was calculated using the formula: (Vmax of test sample—Vmin of test sample)/(Vmax of vehicle—Vmin of vehicle) × 100.

### 2.8. KLK5 Expression

HEKn was plated on a 12-well plate at 1 × 10^5^ cells/well and incubated until 90–100% confluency was reached. HEKn were pretreated with ALE in the presence of 10, 20, and 50 µg/mL or 200 nM isotretinoin for 2 h, followed by treatment with 1,25 (OH)_2_VD_3_ (VD_3_), a biologically active form of vitamin D at 37 °C for 48 h. After centrifugation of the cell culture medium, the supernatants were measured for KLK5 using human kallikrein 5 ELISA kit (R&D systems).

### 2.9. Cathelicidin Cleavage Assay

HEKn was plated on a 12-well plate at 1 × 10^5^ cells/well and incubated until 90–100% confluency was reached. HEKn were pretreated with isochlorogenic acids A and C in the presence of 10, 20 µM or 1 µM leupeptin for 2 h, followed by treatment with VD_3_ at 37 °C for 48 h. After centrifugation of the cell culture medium, the supernatants were collected and lyophilized. The sample was dissolved with the same volume of deionized water and subjected to Tricine-SDS-PAGE (Koma Biotech, Seoul, Korea), Western blot analysis was performed with anti-LL-37 antibody (Novus Biologicals, Centennial, CO, USA).

### 2.10. Co-Culture Conditions

Co-culture experiments were performed using Millicell Cell culture insert (Millipore, Billerica, MA, USA), according to the manufacturer’s instructions.

#### 2.10.1. HEKn/THP-1 Co-Culture

HEKn was seeded onto 0.4-μm-pore transwell inserts at a density of 1 × 10^3^ cells and maintained for 48 h. THP-1 was seeded on 12-well plates at 5 × 10^5^ cells/well and differentiated into macrophages using 100 nM phorbol 12-myristate 13-acetate, PMA (Sigma-Aldrich) for 24 h. The culture media of both cells were replaced with fresh DMEM containing 1% FBS and 1% penicillin/streptomycin (Gibco). Subsequently, transwell inserts were placed in the upper compartment of THP-1-seeded 12-well plates. Upper inserts were pretreated with isochlorogenic acid A or isochlorogenic acid C for 1 h, followed by treatment with VD_3_. After 72 h incubation, the secreted pro-inflammatory cytokines were measured by ELISA (human TNF-α and IL-1β ELISA kit, R&D systems).

#### 2.10.2. HEKn/MC/9 Co-Culture

HEKn was seeded onto 0.4-μm-pore inserts at a density of 1 × 10^3^ cells and maintained for 48 h. The culture medium of HEKn were replaced with fresh DMEM containing 1% FBS and 1% penicillin/streptomycin (Gibco) and pretreated with isochlorogenic acid A or isochlorogenic acid C for 1 h, followed by treatment with VD_3_. They were further cultured for 48 h. MC/9 (5 × 10^6^) was seeded on 12-well plates, and treated with Boc-Gln-D-Trp (Formyl)-Phe benzyl ester trifluoroacetate salt (QWF, R&D systems), a Mas-related G-protein-coupled receptor member X2 (MRGPRX2) antagonist for 1 h. Subsequently, the transwell inserts were placed in the upper compartment of MC/9-seeded 12-well plates. After incubation for 12 h, MC/9 degranulation was assessed by measuring β-hexosaminidase activity and histamine release, and the expression of mast cell protease was measured in MC/9.

### 2.11. Total RNA Extraction, cDNA Synthesis, and Quantitative PCR

Total RNA was extracted using RNeasy kit (Qiagen, Hilden, Germany). The cDNA was synthesized using an AccuPower^®^ CycleScript RT PreMix (Bioneer, Daejeon, Korea), according to the manufacturer’s instructions. The *Cma1* (the gene for Chymase), *Tpsab1* (the gene for tryptase), *ICAM1*, *VCAM1*, and *FPR2* (the gene for FPRL1) mRNAs were measured by quantitative real-time PCR. The primer sequences were shown in Table 1. All mRNA data were normalized to GAPDH expression.

### 2.12. HMEC-1 Proliferation Assay

HMEC-1 proliferation was measured using the water-soluble tetrazolium salt (WST, EZ-Cytox, DoGenBio, Seoul, Korea). Cells were plated in triplicate on the 24-well plate and incubated for 24 h. The cells were pretreated with different isochlorogenic acid A concentrations (1, 10, and 50 µg/mL), followed by PD98059 (Cell Signaling Technology, Danvers, MA, USA), a ERK1/2 inhibitor; SB203580 (Cell Signaling Technology), a p38MAPK inhibitor; or WRW4 (R&D systems), a FPRL1 antagonist; and then treated with 10 µg/mL LL-37 (Carbosynth, Berkshire, UK) under supplement-free conditions. After 72 h of incubation, cells were treated with WST, and further incubated for 2 h. Absorbance was measured at 450 nm via microplate spectrophotometry.

### 2.13. HMEC-1 Migration Assay

HMEC-1 (2 × 10^5^ cells/well) were plated on a 12-well plate and incubated until 90–100% confluency was reached. The cell monolayers were scratched using a sterile pipette tip. The cells were washed with phosphate buffered saline (PBS) to eliminate cell debris and replaced with supplement-free MCDB 131 containing 1% FBS. The cells were pretreated with isochlorogenic acid A or isochlorogenic acid C (1, 10, and 50 µg/mL), followed by treatment with 5 µg/mL LL-37. After 48 h of incubation, microscopic images were captured, and the migration was measured by analyzing the images of cells filling the scratch.

### 2.14. Statistical Analysis

Differences between the control and treatment group were analyzed using Student *t* test; *p* < 0.05 was considered statistically significant.

## 3. Results

### 3.1. ALE Inhibits KLK5 Activity

Individuals at risk of rosacea express not only abnormally high levels of KLK5 in their epidermis, but also higher levels of LL-37, the proteolytically cleaved cathelicidin, compared with normal individuals [6]. The extracellular cathelicidin is activated into the bioactive fragment LL-37 via tightly regulated proteolytic cleavage by KLK5. However, in patients with rosacea, the abnormal activation of cathelicidin into LL-37 due to excessive KLK5 leads to skin inflammation and erythema. KLK5 is known to initiate a cascade reaction via auto-activation, suggesting that inhibiting the expression or activity of KLK5 can control the generation of LL-37 induced by excessive activation of KLK5. Therefore, we evaluated whether ALE extract affected KLK5 expression in HEKn and extracellular enzymatic activity. ALE showed no significant effect on KLK5 expression, but effectively inhibited KLK5 protease activity in extracellular enzymatic reactions using a trypsin-like serine protease-specific fluorogenic peptide substrate, Boc-Val-Pro-Arg-7-amido-4-methylcoumarin hydrochloride (Boc-V-P-R-AMC) (Figure 1A,B). This result confirmed that ALE regulates the transformation of cathelicidin into active LL-37 by regulating KLK5 protease activity, but not KLK5 expression.

### 3.2. Isolation and Identification of Active Compounds from A. lavandulaefolia as KLK5 Inhibitors

To determine the bioactive compounds of ALE inhibiting KLK5 activity, we performed bioassay-guided fractionation via chromatographic separation. The dried aerial part of *A. lavandulaefolia* was extracted with water and the soluble part was lyophilized and separated into six different fractions via microporous resin HP-20 column chromatography. Next, we identified the active compounds in six different fractions inhibiting KLK5 activity: fraction 1 (non-binding flow through), fraction 2 (eluted with 20% ethanol), fraction 3 (eluted with 40% ethanol), fraction 4 (eluted with 60% ethanol), fraction 5 (eluted with 80% ethanol), fraction 6 (eluted with 100% ethanol) (Figure 2A). We found that the fraction 3 (eluted with 40% ethanol) most effectively inhibited KLK5 activity (Appendix A). As a result of HPLC analysis, it was confirmed that fraction 3 showed two major peaks, and these peaks were identified by comparing the HPLC retention time with the UV pattern (Figure 2B). In addition, the chemical compositions of peak 1 and peak 2 were analyzed by using ESI-MS/MS (Figure 2C). The fragmentation patterns of peak 1 showed molecular ions at *m*/*z* 515.1237 [M–H]^−^, 353.0889 [M–C_9_H_7_O_3_–H]^−^, 191.0549 [M–2(C_9_H_7_O_3_)–H]^−^, 179.0338 [M–C_9_H_7_O_3_–C_7_H_12_O_6_–H]^−^, and 135.0436 [M–C_9_H_7_O_3_–C_7_H_12_O_6_–CO_2_–H]^−^ (Figure 2C). Accordingly, peak 1 was identified as 3,5-dicaffeoylquinic acid (isochlorogenic acid A) (Figure 2C). Peak 2 showed molecular ions at *m*/*z* 515.1238 [M–H]^−^, 353.0895 [M–C_9_H_7_O_3_–H]^−^, 191.0553 [M–2(C_9_H_7_O_3_)–H]^−^, 179.0341 [M–C_9_H_7_O_3_–C_7_H_12_O_6_–H]^−^, 173.0445 [M–2(C_9_H_7_O_3_)–H_2_O–H]^−^,and 135.0441 [M–C_9_H_7_O_3_–C_7_H_12_O_6_–CO_2_–H]^−^ (Figure 2C). The fragmentation pattern of peak 2 was confirmed to be similar to peak 1. Therefore, peak 2 was unequivocally distinguished by forming an ion at *m*/*z* 173.5 of the MS/MS spectrum. As a result, peak 2 was confirmed to be 4,5-dicaffeoylquinic acid (isochlorogenic acid C) (Figure 2C). To investigate whether identified two compounds effectively inhibited KLK5 activity, the inhibitory effects of isochlorogenic acids A and C on KLK5 activity were measured via extracellular enzymatic reaction. Both isochlorogenic acids A and C inhibited KLK5 activity in a concentration-dependent manner, and both compounds showed similar inhibitory effect against KLK5 activity (Figure 2D).

### 3.3. Chlorogenic Acid Isomers Inhibit the Proteolytic Cleavage of Cathelicidin by Blocking KLK5 Protease Activity

To investigate the inhibitory effects of isochlorogenic acids A and C on the activation of cathelicidin into LL-37 induced by VD_3_, HEKn were pretreated with these compounds, followed by incubation with VD_3_ for 48 h. Activated LL-37 was analyzed via western blot in HEKn culture medium. The results suggested that pro-cathelicidin was hardly induced in the vehicle-treated condition, whereas cathelicidin and activated LL-37 were both detected following VD_3_ treatment (Figure 3). Pro-cathelicidin was cleaved into LL-37 by activated KLK5, indicating that proteolytic cleavage was inhibited by isochlorogenic acids A and C, and leupeptin, a specific serine protease inhibitor (Figure 3). These results indicate that isochlorogenic acids A and C inhibited the activation of the precursor cathelicidin secreted by HEKn into LL-37, a biologically active form by regulating the proteolytic activity of KLK5.

### 3.4. Chlorogenic Acid Isomers Decrease the Expression of Proinflammatory Cytokines Induced by VD_3_

LL-37 is known to induce an inflammatory response in rosacea lesions by increasing the synthesis of inflammatory mediators. In our previous study, VD_3_ increased the activity of KLK5 in HEKn, leading to proteolytic activation of cathelicidin into LL-37, but the expression of pro-inflammatory cytokines was hardly induced by LL-37 in HEKn cultured alone [27]. Therefore, a model of rosacea inflammation was constructed to demonstrate the stimulation of THP-1 by LL-37 induced by VD_3_ in HEKn to generate pro-inflammatory cytokines under HEKn/THP-1 co-culture. Isochlorogenic acids A and C inhibited KLK5 activity without a significant inhibitory effect of the expression of pro-inflammatory cytokines induced by LL-37 under THP-1 monoculture (Appendix A). In contrast, the chlorogenic acid isomers A and C exhibited significant inhibitory effects on the expression of proinflammatory cytokines induced by VD_3_ in HEKn/THP-1 co-culture conditions, and showed almost similar efficacy (Figure 4). Taken together, these results suggest that isochlorogenic acids A and C regulate the inflammatory response by inhibiting the activation of cathelicidin in HEKn, rather than suppressing the expression of proinflammatory cytokines by directly targeting THP-1.

### 3.5. Inhibition of VD_3_-Induced Mast Cell Activation by Chlorogenic Acid Isomers

Mast cells are the major mediators induced by LL-37 during inflammation and erythema in rosacea [17]. Mast cell activation triggers the release of preformed mediators stored within mast cell secretory granules, including histamine and proteases, followed by the production of prostaglandins, chemokines, cytokines, and growth factors [28]. The major proteases of mast cell-secreting granules include chymase, tryptase, and carboxypeptidase A3, which are specifically expressed in mast cells and are stored in the active form in the granules [29,30]. Recent studies investigating the role of mast cells in inflammation have focused on inhibiting the release of specific chymase and tryptases from mast cells [13]. Mast cell activation and degranulation are major factors determining the pathological severity of rosacea [31]. To establish the response of mast cell degranulation to direct LL-37 stimulation induced by VD_3_ in keratinocytes, MC/9 cells were co-cultured with HEKn treated with isochlorogenic acids A and C. HEKn were preincubated with isochlorogenic acids A and C for 1 h, and then stimulated with VD_3_. β-hexosaminidase activity and histamine release were investigated as markers of mast cell degranulation. Histamine release and β-hexosaminidase activity were increased by VD_3_ under co-culture conditions, but not in the absence of VD_3_ treatment (Figure 5A). Histamine release and β-hexosaminidase activities are inhibited by isochlorogenic acids A and C (Figure 5A). The inhibitory effects were similar to those observed following treatment with leupeptin, a KLK5 inhibitor and QWF, a MRGPRX2 antagonist (Figure 5A). In addition, to investigate whether isochlorogenic acids A and C affect mast cell activation, mRNA expression of chymase and tryptase was measured in VD_3_-induced mast cell activation. The levels of both chymase (*Cma1*) and tryptase (*tpsab1*) mRNA were increased by VD_3_, and these increases were suppressed by isochlorogenic acids A and C (Figure 5B). Based on these results, isochlorogenic acids A and C may affect mast cell degranulation by inhibiting cathelicidin activation via inhibition of KLK5 activity. In addition, it was established that mast cells stimulated by LL-37 were activated via MRGPRX2-mediated signaling pathway.

### 3.6. Isochlorogenic Acid Isomers Regulate LL-37-Induced Vascular Endothelial Cell Activation

In rosacea, LL-37 enhances the proliferation of vascular endothelial cells and induces angiogenesis, leading to erythema [4]. Therefore, we first investigated the effects of isochlorogenic acids A and C on the proliferation and migration of HMEC-1 initiated by LL-37. HMEC-1 proliferation and migration were increased by LL-37, which was effectively inhibited by these isoforms. It is known that LL-37 exhibits proliferation, angiogenesis, and immunomodulatory activity via MAPK pathway in various cell lines by activating their major receptor, FPRL-1 [3,4]. The FPRL1-mediated MAPK signaling pathway in the LL-37-induced activation of HMEC-1 was inhibited by WRW4, a selective FPRL1 inhibitor, along with PD98059, a selective ERK1/2 inhibitor, and SP203580, a selective p38MAPK inhibitor, thereby affecting the proliferation and migration of HMEC-1. Inhibition of FPRL-1, ERK1/2, and p38MAPK affects the proliferation of HMEC-1 (Figure 6A). In addition, the expression of FPRL-1 mRNA was measured in LL-37-induced HMEC-1 and showed no significant change (Appendix A). These results indicate that LL-37-stimulated HMEC-1 activates proliferation via FPRL-1 mediated MAPK signaling regulated by isochlorogenic acids A and C (Figure 6A). Recent studies have shown that endothelial adhesion molecules such as VCAM-1 and E-selectin play a role in endothelial cell growth, and ICAM-1 is known to control the motility of vascular endothelial cells migrating during angiogenesis [32,33]. In this study, the isochlorogenic acids A and C inhibited the mRNA expression of adhesion molecules, ICAM-1 and VCAM-1 induced by LL-37 in HMEC-1 (Figure 6B). Additionally, both isochlorogenic acids A and C significantly inhibited the migration of HMEC-1 induced by LL-37 (Figure 6C). These results indicate that both isochlorogenic acids A and C inhibit LL-37-induced MAPK signal transduction via FPRL1, a specific receptor for LL-37 in HMEC-1, and also inhibit the mRNA expression of adhesion molecules, ICAM-1, and VCAM-1, thereby effectively regulating proliferation and migration in LL-37-induced angiogenesis.

## 4. Discussion

Cathelicidin expression correlates with effective innate immune defense of the skin in chronic inflammatory diseases of skin [34]. In psoriatic skin lesions, the expression of LL-37, the active form of cathelicidin, is upregulated, and secondary infection is rare. In contrast, in atopic dermatitis, the expression of LL-37 and other antibacterial peptides is downregulated, which leads to skin inflammation [35]. In addition, an enhanced innate immune response due to aberrant expression of cathelicidin has been shown to be a major contributor to the pathophysiology of rosacea. A recent study showed that the expression of dermal serine proteases, KLK5 and cathelicidin, was significantly elevated in rosacea lesions compared with healthy skin [6]. The production of KLK5 and cathelicidin and consequent activation of cathelicidin into LL-37 by KLK5 may have a major impact on the development of rosacea-associated erythema and inflammation [6].

Here, we confirmed that ALE inhibited the KLK5 protease activity. The two chlorogenic acid isomers, 3,5-dicaffeylquinic acid (isochlorogenic acid A) and 4,5-dicaffeylquinic acid (isochlorogenic acid C) isolated from ALE, were the main ingredients inhibiting KLK5 activity. These results suggest that the isolated chlorogenic acid isomers effectively modulate inflammation and erythema observed in the pathophysiology of rosacea. Interestingly, however, our study confirmed that chlorogenic acid did not affect the proteolytic activity of KLK5 (Appendix A).

*Artemisia* is a medicinally important perennial herb belonging to the Asteraceae family, which includes various species of plants such as *A. annua*, *A. lavandulaefolia*, *A. capillaris*, and *A. princeps*. *Artemisia* species exhibit anti-inflammatory, anti-tumor, anti-rheumatic, and antibacterial activities [36]. *A. annua* is widely used as an antimalarial agent, and artemisinin isolated from *A. annua* exhibits antimalarial activity, and the mechanism of action against malaria was also elucidated [37]. A recent study reported that artemisinin alleviated rosacea symptoms by inhibiting the inflammatory response and angiogenesis induced by LL-37 [38]. Artemisinin is a component commonly found in diverse Artemisia species [39]. The crude ALE used in this study inhibited KLK5 activity; however, artemisinin was not a major component identified in HPLC analysis (Appendix A). Based on these results, we performed bioactivity-guided fractionation to isolate novel active ingredients including two chlorogenic acid isomers, isochlorogenic acid A and isochlorogenic acid C exhibiting inhibitory effects against KLK5 activity. Isochlorogenic acid is known to exhibit antiviral, anti-inflammatory and antioxidant effects [40,41,42]; however, its effect on rosacea symptoms and its mechanism of action have yet to be reported. The isochlorogenic acids A and C isolated from *A. lavandulaefolia* inhibit the proteolytic cleavage of cathelicidin into LL-37 by directly targeting KLK5. Another chlorogenic acid isomer, isochlorogenic acid B, also inhibited KLK5 activity (Appendix A). Chlorogenic acid has been reported to show various biological effects such as anti-inflammatory, antibacterial, anti-acne, antioxidant, anti-obesity, and anti-viral efficacy [43,44]. Interestingly, in our study, the chlorogenic acid isomers inhibited KLK5 activity, whereas chlorogenic acid did not significantly inhibit KLK5 activity (Appendix A). These results suggest that the specific structure of the chlorogenic acid isomers affects the KLK5 activity. In addition, the Lineweaver–Burk plot analysis indicates that isochlorogenic acids A and C act as non-competitive inhibitors of KLK5 (Appendix A). It is expected that chlorogenic acids A and C bind to the allosteric site of KLK5 and act as non-competitive modulators. Structure-activity relationship studies may further clarify the relationship between the chemical structure-related properties of compounds and their biological activity.

The pathophysiology of rosacea is not fully understood and involves a complex interaction between genetic predisposition, UV radiation, exposure to microbes and mites, damaged skin barrier, nerve and vascular dysfunction, and disruption of the immune system [7]. Among them, UV is a major factor causing rosacea, and in rosacea patients with impaired skin homeostasis, UV-generated vitamin D_3_ triggers abnormal immune response and vascular abnormalities in the skin [35]. Therefore, in this study, we investigated the inflammation and erythema induced by VD_3_, an active form of vitamin D_3_ occurring in the skin of rosacea patients. In our previous study, VD_3_ acted as an inflammatory factor, upregulating both KLK5 and cathelicidin expression in HEKn, whereas only a weak cathelicidin expression was induced in THP-1 immune cells [27]. Pro-inflammatory cytokines were hardly induced by VD_3_ in monocultures of HEKn and THP-1. In addition, LL-37 significantly induced the synthesis of proinflammatory cytokines in THP-1, but not in HEKn. Therefore, an in vitro experimental model mimicking skin inflammation in VD_3_-induced rosacea was constructed using a co-culture system of HEKn and THP-1. Isochlorogenic acids A and C did not directly inhibit the expression of proinflammatory cytokines in THP-1 induced by LL-37 (Appendix A). In vitro co-culture studies revealed that isochlorogenic acids A and C inhibit pro-inflammatory cytokines by blocking the generation of LL-37 via inhibition of KLK5 activity.

Activated mast cells secrete various immune inflammatory mediators, including histamine, tryptase, and chymase, which are considered major factors causing inflammation of rosacea [17]. Since LL-37 represents a major trigger in mast cell activation, the inhibition of LL-37-mediated mast cell activation is a major target to control inflammation in rosacea. Similar to the macrophage-mediated inflammatory response, VD_3_ did not directly induce the activation of mast cells, MC/9. Therefore, mast cells were activated via interaction with HEKn using the HEKn/MC/9 co-culture system. Isochlorogenic acids A and C did not directly inhibit mast cell degranulation induced by LL-37, whereas VD_3_-induced mast cell degranulation was inhibited under co-culture conditions. This result is speculated to be an effect of the inhibition of LL-37 generation mediated via inhibition of KLK5 activity by isochlorogenic acids A and C. Recently, MRGPRX2, a Mas-related G protein-coupled receptor primarily expressed in mast cells, was identified as a receptor for LL-37 in mast cell activation [45]. Therefore, the role of LL-37 as an inducer of mast cell activation was established via inhibition of MRGPRX2-mediated signaling using the MRGPRX2 antagonist. Treatment of mast cells with QWF, a specific antagonist of MRGPRX2, resulted in a significant inhibition of the release of histamine induced by VD_3_. These results demonstrate that LL37/MRGPRX2 signaling is the major pathway in VD_3_-induced mast cell activation.

Mast cells are localized in the dermis close to nerve endings and blood vessels involved in wound healing and immune response. Therefore, activated mast cells attract other immune mediators through the production of antimicrobial peptides, inflammatory cytokines, chemokines, and proteases, leading to vasodilation and angiogenesis [46,47,48]. Isochlorogenic acids A and C inhibit the mRNA expression of mast cell proteases, chymase and tryptase, in addition to exhibiting inhibitory effects on the proliferation and migration of vascular endothelial cells induced by the activated antibacterial peptide, LL-37. Further, the LL-37-induced proliferation of vascular endothelial cells was inhibited by treatment with LL-37 receptor (FPRL1) antagonist, WRW4; p38MAPK inhibitor, SB203580, and; ERK inhibitor, PD98059. These results indicate that LL-37-induced proliferation of vascular endothelial cells is mediated via the FPRL1-mediated MAPK signaling pathway. Isochlorogenic acids A and C not only inhibited the production of mediators such as LL-37 leading to angiogenesis, but also blocked angiogenesis by directly targeting the activation of vascular endothelial cells induced by activated LL-37.

This study revealed, for the first time, that isochlorogenic acids A and C isolated from *A. lavandulaefolia* regulate the activation of cathelicidin to LL-37 by inhibiting the activity of KLK5. In addition, the inhibition of LL-37 activation by isochlorogenic acids A and C regulates the function of immune cells and blood vessels in rosacea. The mechanism of action underlying the inhibition of activation signals was also elucidated (Figure 7). Although this study is a preliminary investigation into the effect of chlorogenic acid isomers on immunomodulation and erythema improvement in rosacea lesions, it demonstrates their potential therapeutic efficacy in rosacea management. The evaluation of therapeutic efficacy of chlorogenic acid isomers in rosacea requires further clinical studies.

## Figures and Tables

**Figure 1 biomedicines-10-00463-f001:**
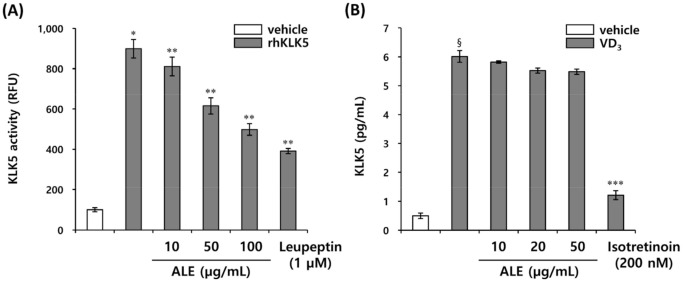
ALE inhibits KLK5 protease activity. (**A**) KLK5 activity was measured in relative fluorescence units (RFUs) using Boc-V-P-R-AMC, fluorogenic peptide substrate. (**B**) KLK5 expression was measured in HEKn induced by VD_3_. The results are expressed as mean ± standard deviation (SD) (*n* = 3). * *p* < 0.01 vs. vehicle control; ** *p* < 0.05 vs. rhKLK5-treated control; § *p* < 0.01 vs. VD_3_-untreated control; *** *p* < 0.05 vs. VD_3_-treated control.

**Figure 2 biomedicines-10-00463-f002:**
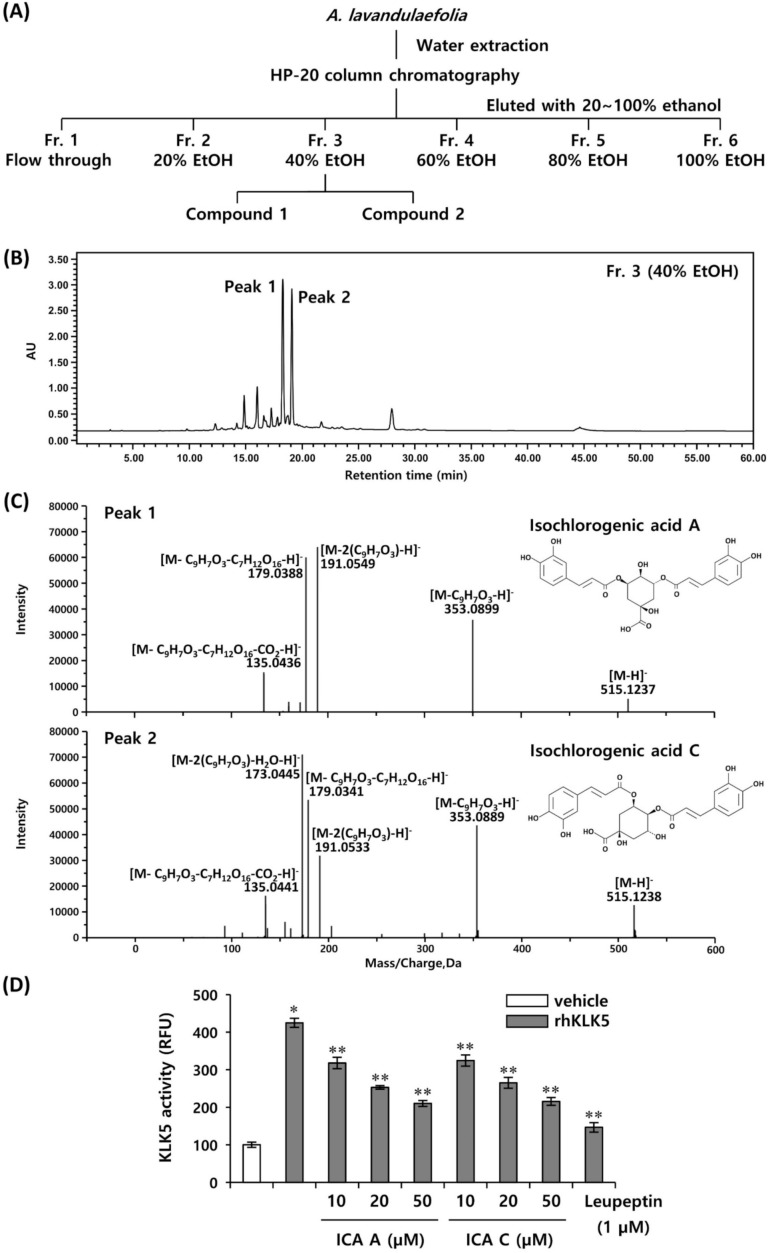
Bioassay-guided isolation and identification of bioactive compounds from *A. lavandulaefolia* inhibiting KLK5 activity. (**A**) Schematic representation of bioassay-guided isolation of bioactive compounds. (**B**) HPLC chromatogram of fractionation 3. (**C**) Negative ion MS/MS spectra for compounds 1 and 2. (**D**) The inhibitory effect of compounds 1 and 2 on KLK5 activity was measured using Boc-V-P-R-AMC, fluorogenic peptide substrate. The results are expressed as mean ± standard deviation (SD) (*n* = 3). * *p* < 0.01 vs. vehicle control; ** *p* < 0.01 vs. rhKLK5-treated control. ICA A, isochlorogenic acid A; ICA C, isochlorogenic acid C.

**Figure 3 biomedicines-10-00463-f003:**
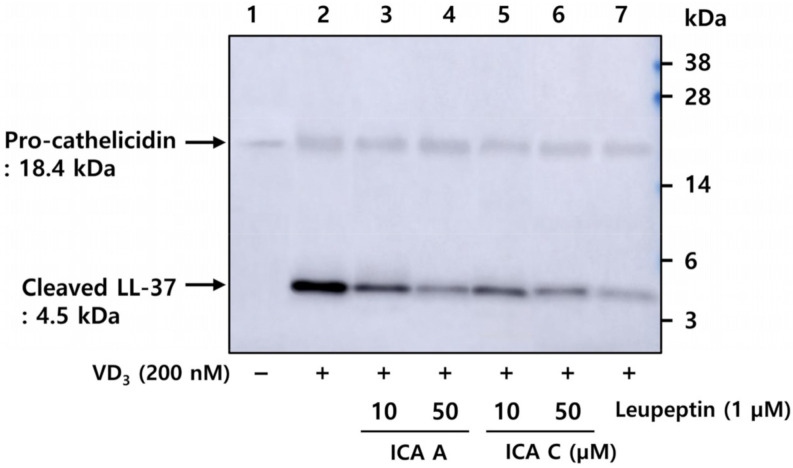
Chlorogenic acid isomers inhibit the proteolytic cleavage of cathelicidin by blocking KLK5 activation. HEKn was pretreated with 10 and 20 µM isochlorogenic acid A and isochlorogenic acid C, respectively, followed by 1 µM leupeptin, a serine protease inhibitor, for 1 h, and subsequently treated with 200 nM VD_3_ for 48 h. HEKn culture media were collected and subjected to Western Blot analysis with anti-LL-37 antibody. Lane 1, DMSO vehicle control; lane 2, HEKn + VD_3_; lanes 3 and 4, HEKn + VD_3_ + 10, 50 µM isochlorogenic acid A; lane 5 and 6, HEKn + VD_3_ + 10, 50 µM isochlorogenic acid C; lane 7, HEKn + VD_3_ + 1 µM leupeptin. ICA A, isochlorogenic acid A; ICA C, isochlorogenic acid C.

**Figure 4 biomedicines-10-00463-f004:**
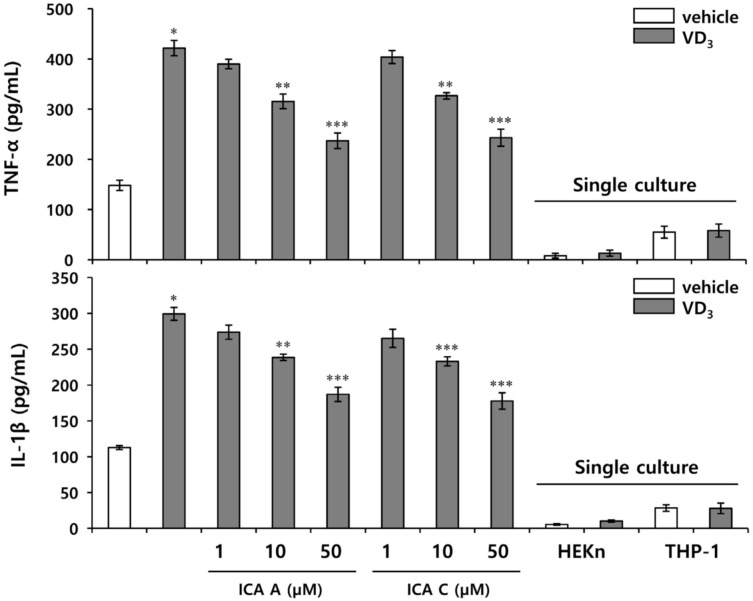
Chlorogenic acid isomers inhibit the expression of proinflammatory cytokines induced by VD_3_ in HEKn/THP-1 co-culture. HEKn were seeded on upper insert chambers, and THP-1 was seeded on lower chamber in the two-compartment transwell system for 24 h. Upper insert was pretreated with 1, 10, or 50 µM isochlorogenic acids A and isochlorogenic acid C, respectively, for 1 h, followed by treatment with 200 nM VD_3_. After 72 h incubation, the secreted pro-inflammatory cytokines (TNF-α, IL-1β) were measured by ELISA. The results are expressed as mean ± standard deviation (SD) (*n* = 3). * *p* < 0.01 vs. vehicle control; ** *p* < 0.05 vs. VD_3_-treated control; *** *p* < 0.01 vs. VD_3_-treated control.

**Figure 5 biomedicines-10-00463-f005:**
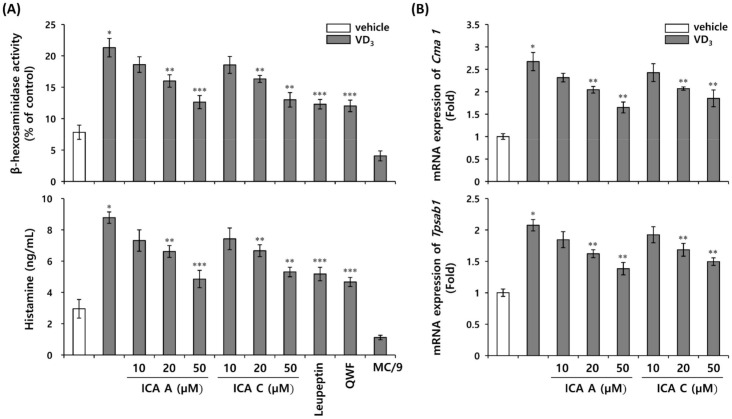
Chlorogenic acid isomers inhibit mast cell activation induced by VD_3_ in HEKn/MC/9 co-culture system. HEKn were seeded onto upper insert chambers, and MC/9 cells on the lower chamber of the two-compartment transwell system for 24 h. The upper insert chamber was pretreated with 1, 10, or 50 µM isochlorogenic acids A and isochlorogenic acid C, and 1 µM leupeptin for 1 h, and subsequently treated with 200 nM VD_3_. Lower chamber was treated with 10 µM QWF, an MRGPRX2 antagonist. (**A**) Mast cell degranulation was measured by monitoring the release of β-hexosaminidase and histamine. (**B**) The expression of mast cell protease, chymase (*Cma1*) and tryptase (*Tpsab1*) was analyzed via quantitative real-time PCR. The results are expressed as mean ± standard deviation (SD) (*n* = 3). * *p* < 0.01 vs. vehicle control; ** *p* < 0.05 vs. VD_3_-treated control; *** *p* < 0.01 vs. VD_3_-treated control. ICA A, isochlorogenic acid A; ICA C, isochlorogenic acid C.

**Figure 6 biomedicines-10-00463-f006:**
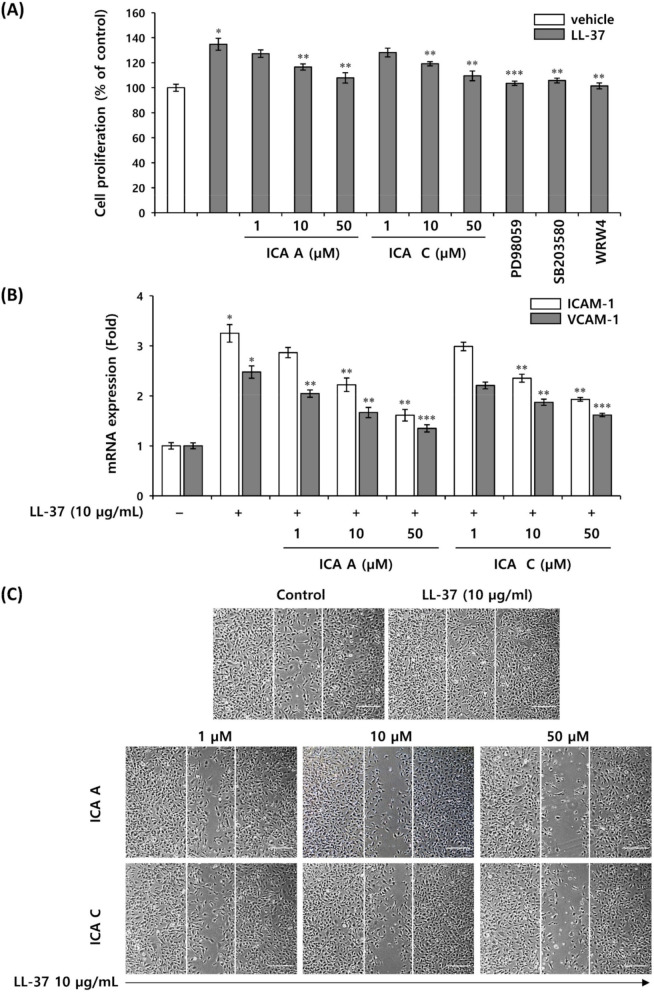
Chlorogenic acid isomers inhibit vascular endothelial cell activation. HMEC-1 was pretreated with isochlorogenic acids A, isochlorogenic acid C, 20 µM ERK1/2 inhibitor, 10 µM p38MAPK inhibitor, and 10 µM FPRL1 antagonist for 1 h, and subsequently treated with 5 µg/mL LL-37. (**A**) Cell proliferation was measured using the WST1 assay. (**B**) The expression of cell adhesion molecules, ICAM-1 and VCAM-1 mRNA was measured by quantitative real-time PCR. (**C**) Cell migration was measured using scratch migration assay. Representative image of the inhibition of cell migration by isochlorogenic acids and C, compared with vehicle-treated control; scale bars are 100 µm. The results are expressed as mean ± standard deviation (SD) (*n* = 3). * *p* < 0.01 vs. vehicle control; ** *p* < 0.05 vs. LL-37-treated control; *** *p* < 0.01 vs. LL-37-treated control. ICA A, isochlorogenic acid A; ICA C, isochlorogenic acid C.

**Figure 7 biomedicines-10-00463-f007:**
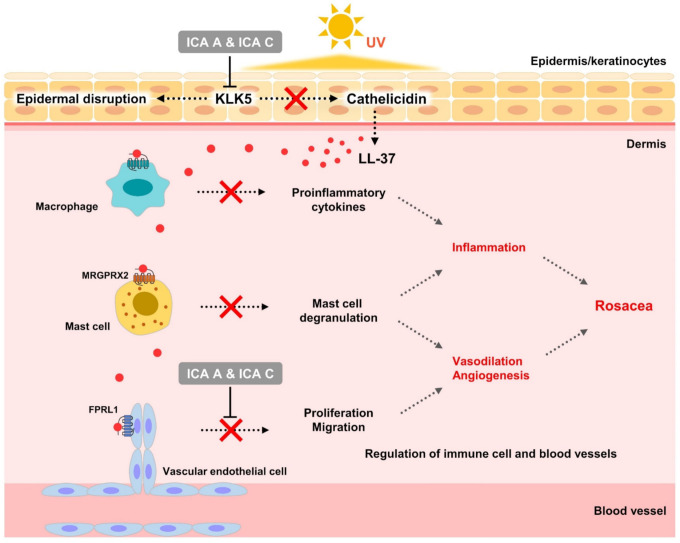
Mechanism of action of isochlorogenic acids A and C in rosacea. ICA A, isochlorogenic acid A; ICA C, isochlorogenic acid C.

**Table 1 biomedicines-10-00463-t001:** Primer sequence for PCR.

Gene	Primer Sequence (5′-3′)	Product Size (bp)
*Cma1*	Forward TCTGCTGTGTGCTGGGATAG	190
	Reverse GGCACACAAAACCTGCACTA	
*Tpsab1*	Forward GTGCTGGGAATGAAGGACAT	187
	Reverse TTGGGGACATAGTGGTGGAT	
*ICAM1*	Forward GGCTGGAGCTGTTTGAGAAC	202
	Reverse ACTGTGGGGTTCAACCTCTG	
*VCAM1*	Forward CAGACAGGAAGTCCCTGGAA	212
	Reverse TTCTTGCAGCTTTGTGGATG	
*FPR2*	Forward CAACCCCATGCTTTACGTCT	184
	Reverse ATATCCCTGACCCCATCCTC	
*Mrgprx2*	Forward TGAAAGCAACCATACTGGAATGTC	113
	Reverse ACCACAGCACTGTGGCATTTCC	
*GAPDH*	Forward TGCACCACCAACTGCTTAGC	87
	Reverse GGCATGGACTGTGGTCATGAG	
*Gapdh*	Forward CATCACTGCCACCCAGAAGACTG	153
	Reverse ATGCCAGTGAGCTTCCCGTTCAG	

*Note*: *Cma1*, mouse chymase; *Tpsab1*, mouse tryptase; *ICAM1*, human intercellular adhesion molecule 1; *VCAM1*, human vascular cell adhesion molecule 1; *FPR2*, human formyl peptide receptor 2; *Mrgprx2*, mouse MAS-related G protein-coupled receptor member X2; *GAPDH*, human glyceraldehyde-3-phosphate dehydrogenase; *Gapdh*, mouse glyceraldehyde-3-phosphate dehydrogenase.

## Data Availability

The data presented in this study are available on request from the corresponding author.

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
