# Peer review of "Chlorogenic Acid Isomers Isolated from Artemisia lavandulaefolia Exhibit Anti-Rosacea Effects In Vitro"

_biomedicines, 2022, doi:10.3390/biomedicines10020463_

Round 1

Reviewer 1 Report

Roh et al. examined the effects of herbal extracts from perennial used in traditional medicine. They essentially attested whether the extracts work similarly as an antimalarial drug [artemisinin derived from Artemisia annua L. did (ref. 34)].

Instead, the authors used Artemisia lavandulaefolia and identified active polyphenol compounds different from artemisinin: the isomers of chlorogenic acid.

This manuscript is well designed/written and could provide substantial novelty to the fields of cutaneous innate immunity.

Here are the comments.

1. A previous work demonstrated artemisinin's inhibitory effects on the NFkB pathway. The addition of such data would improve the quality even more.

2. Although they speculated that chlorogenic acids A/C interfere with KLK5's allosteric sites (L441~), the data presented in Fig. S4 lacks positive control, such as the broad protease inhibitor leupeptin or negative control nonisomeric chlorogenic acids.  

3. Most readers are not knowledgeable about herbal extracts (polyphenols such as tannin). Authors may add general and easy-to-understand information on such natural antioxidants to the beginning of the introduction/discussion sections. 

Reviewer 2 Report

Introduction: improve Artemisia lavandulaefolia effects section particularly, the authors should prove efficacy properties based on the literature

Methods: I suggest performing a pilot experiment using an in vivo model of rosacea and then reporting skin irritation parameters (erythema, edema, exfoliation etc) to better confirm in vitro results. It could be even better, to evaluate cytokines expression also in vivo 

Reviewer 3 Report

Authors wrought that chlorogenic acid A and C inhibit KLK5 activity and gene expression induced by LL-37 in vitro. KLK5 and LL-37 are key molecules against to treatment of rosacea. This paper is simple, but I have many strange points in this paper.

major comments

Authors show same graphs in this paper, but I didn’t recognized number of experiment each graphs. It is very important to analyze and summarize data based on statistics.

Authors shows that ALE inhibits KLK5 activity but not expression of KLK5 in Figure 1, and wrought ALE showed no significant effect on KLK5 expression, but Figure 1 B shows KLK5 expression induced by VD3 in HEKn, right? In addition, authors don’t show how to measure KLK5 expression and amount of KLK5 protein. In Figure 1 A, I couldn’t understand means of vehicle. Probably, I think other recombinant protein are used against to recombinant human KLK5 protein, in this case you should use “control”. I wonder why these results are showed in Figure 1. Is it need? In addition, section 3.1 contains a lot of discussion. This is the results section, please show only the results.

It is indicated that isochlorogenic acid A, B and C were purchased from the company in materials and methods section. If we can obtained these compounds from a company, we don’t need preparation ICA from A. lavendulaefolin. Therefore, it isn’t need this section. If you used for standard of HPLC, you have to wright that.

In materials and methods section, you indicate to use C18(2) column for preparation of ICAs, but in Figure 2 A, is indicated HP-20 column. You use both column or not ?

In Figure 4, you show the results of amount of TNF-alpha and IL-1beta from HEKn/THP-1 co-culture induced by VD3. Figure legend starts that TNF-alpha and IL-1 beta were measured by ELISA, but not materials and methods section. What company and what did you use?

Authors suggested that ICA A and C inhibited cell proliferation of HMEC-1 by inhibiting of phosphorylation of p38 and p42/44 in Figure 6 A. However, you don’t show inhibition of phosphorylation of p38 and p42/44 using western blot or FCM analysis directly. You only compared with inhibitors of p38 and p42/44. This is not evidence of inhibition of the MAPK signal transduction. Is the cell proliferation inhibitory effect of ICA specific to HMEC-1? How about keratinocyte, fibroblast or other immune cells?

What is it the benefit to use ICA A and C for treatment of rosacea? stability, specificity, low IC50, low toxic effect, high permeability and so on. You showed leupeptin is effective rather than ICA A and C in Figure 1B or Figure 5.

This paper is childish, inconsistent, and has many contradictions. Therefore, I think that reject is appropriate.

Reviewer 4 Report

This is an interesting paper, which investigates promising new methods for treatment of rosacea, which is notoriously chronic and difficult to treat.

The study is well performed and well written.

Round 2

Reviewer 1 Report

No comments.